# Scenarios of Genes-to-Terpenoids Network Led to the Identification of a Novel *α/β*-Farnesene/*β*-Ocimene Synthase in *Camellia sinensis*

**DOI:** 10.3390/ijms21020655

**Published:** 2020-01-19

**Authors:** Jieyang Jin, Shangrui Zhang, Mingyue Zhao, Tingting Jing, Na Zhang, Jingming Wang, Bin Wu, Chuankui Song

**Affiliations:** State Key Laboratory of Tea Plant Biology and Utilization, International Joint Laboratory on Tea Chemistry and Health Effects, Anhui Agricultural University, Hefei 230036, China; jjyjieyang@163.com (J.J.); spongebob41@163.com (S.Z.); 18656071627@163.com (M.Z.); jingtingting@ahau.edu.cn (T.J.); 15038327156@163.com (N.Z.); wangjm0611@163.com (J.W.); 19840176923@163.com (B.W.)

**Keywords:** terpene synthase, metabolic profiling, gene-to-terpene network, farnesene, β-ocimene, gene silencing, *Camellia sinensis*

## Abstract

Terpenoids play vital roles in tea aroma quality and plants defense performance determination, whereas the scenarios of genes to metabolites of terpenes pathway remain uninvestigated in tea plants. Here, we report the use of an integrated approach combining metabolites, target gene transcripts and function analyses to reveal a gene-to-terpene network in tea plants. Forty-one terpenes including 26 monoterpenes, 14 sesquiterpenes and one triterpene were detected and 82 terpenes related genes were identified from five tissues of tea plants. Pearson correlation analysis resulted in genes to metabolites network. One terpene synthases whose expression positively correlated with farnesene were selected and its function was confirmed involved in the biosynthesis of *α*-farnesene, *β*-ocimene and *β*-farnesene, a very important and conserved alarm pheromone in response to aphids by both in vitro enzymatic assay in planta function analysis. In summary, we provided the first reliable gene-to-terpene network for novel genes discovery.

## 1. Introduction

Tea infusions made from *Camellia sinensis* are among the most consumed beverages around the world. The popularity of tea as a global beverage is due to its pleasant flavor and health benefits. Terpenoids belong to the largest class of natural compounds and have a significant influence on the formation of tea aroma due to their low odor thresholds and pleasant fragrances [1]. In semi-fermented and fermented teas, linalool and linalool oxides diastereomer are present in concentrations of up to 50% of the volatiles [2] and offer different flavor characteristics. Linalool and sesquiterpene nerolidol are potent odorants of black tea [3], as they impart tea products with a creamy, floral odor and have a low threshold of perception [3]. Both linalool and nerolidol are detectable in tea floral scent [4] and accumulate as non-odorous glycosides in fresh tea leaves [5].

Apart from the odorant properties, terpenoid volatiles also act as chemical messengers or attractants for pollinators [6], deterrents of herbivore predators [7], plant–plant signals [8] and other ecological function [9]. (*E*)-Nerolidol and its derivative 4,8-dimethylnona-1,3,7-triene (DMNT) are vital for plants in response to herbivore attack [10]. Sesquiterpene farnesene was reported as an important and conserved alarm pheromone in response to a variety of aphids species, a major pest in agriculture, which seriously occurs in tea gardens in some countries that produce a great number of merchant tea [11,12]. Some attempts have been made to use synomones and kairomones in the manipulation of natural enemies to suppress pests [13]. Interestingly, several plants succeed in keeping away aphids by synthesizing farnesene, thus indicating that farnesene can be efficiently utilized as an aphid repellent [14].

Terpenoids are biosynthesized via two pathways: the cytosolic mevalonic acid (MVA) pathway and the plastidic methylerythritol phosphate (MEP) pathway (Figure 1). Both pathways lead to the formation of the C5 compounds isopentenyl pyrophosphate (IPP) and dimethylallyl pyrophosphate (DMAPP). In the MEP pathway, IPP is synthesized from pyruvate and glyceraldehyde -3-phosphate [15], while in the cytosol, IPP is derived from acetyl-CoA [16]. The C5 compounds for monoterpene synthesis are derived from the plastidic pathway. Geranyl pyrophosphate (GPP) is the precursor for monoterpenoids, and specific enzymes involved in the biosynthesis of monoterpenoids have been identified and characterized in various plants. Terpene synthases (TPSs) in plants are extensively studied and have been divided into seven subfamilies. The biosynthesis of terpenoids in *Camellia sinensis* is predicted based on the knowledge of the pathways elucidated in other plants, whereas the scenarios of genes to metabolites of terpenes biosynthesis pathway in tea plants remain uninvestigated and most of the TPSs in tea plants, until now, are unknown, except the recently characterized TPSs involved in the formation of (*E*)-nerolidol, linalool [17,18], ocimene [19] and *α*-farnesene [20].

To characterize terpene biosynthesis, here we report the use of an integrated approach combining metabolites, target gene transcripts and gene function analyses to reveal a gene-to-terpene scenarios network in tea plants. Terpene synthases whose expression positively correlated with farnesene were selected and both in vitro and in planta function analysis confirmed that TPS2 involved in the biosynthesis of *α*-farnesene, *β*-ocimene, as well as *β*-farnesene, a very important and conserved alarm pheromone in response to aphids, in tea plants, suggested the gene-to-terpenes network was a vigorous and reliable method for the discovery of novel genes in terpene pathways.

## 2. Results and Discussion

### 2.1. GC–MS-Based Metabolite Profiles

To understand the volatile compound composition of *C. sinensis*, the volatiles in different tissues including, the bud, the first leaf (FL), the second leaf (SL), mature leaf (ML) and stem of tea plants were examined using gas chromatography-mass spectrometry (GC–MS). As shown in Table 1 and Table 2, 41 terpenoids including, 26 monoterpenes (17 acyclic, six monocyclic and three bicyclic monoterpenes), 14 sesquiterpenes (three acyclic, five monocyclic, four bicyclic and two tricyclic) and one triterpene squalene were identified. Of the 41 identified compounds, 25 were verified based on the authentic standards, the others were identified via their retention index constructed by even-numbered chain n-alkenes (C10–C40) and mass spectrum. The metabolites were differentially produced by stem and leaves of different developing stages, and some metabolites can only be detected in certain tissues at specific positions.

The terpenoid metabolites separated by GC–MS were further used for principle component analysis (PCA) and clustering analysis. It can be seen from the PCA plot (Appendix A) that the accumulation patterns of metabolites from five different tissues could be separated. Moreover, a heatmap coupled with clustering analysis was generated to further understand metabolic differentiation in the five tissues (Figure 2). The compounds from bud and FL were clustered together, closer to SL, and separated from the stem and ML, which was consistent with the PCA results indicated that the patterns of metabolite profiles are affected by the tissue and developmental stage in the tea plants. In most cases, the tissue accumulated terpenes are consistent with the expression of the *TPS* gene [21]. According to the heatmap, farnesene exists in the bud and first leaves, consistent with aphids attack.

### 2.2. Transcriptome Sequencing

To achieve a global overview of the gene expression profile in *C. sinensis*, cDNA libraries of five tissues were constructed and sequenced. Pyrosequencing of five different tissues generated 148075495 reads, including 36293915 from the bud, 25938680 from FL, 28732060 from ML, 28438538 from SL and 28672302 from the stem Appendix A. All sequences were assembled to obtain 1447580 contigs.

According to the functional annotation, 82 genes involved in terpenes pathway have been identified, annotated as 28 enzymes in two known terpene pathways MVA and MEP [15,16] (Figure 1). Heatmaps constructed using 82 genes (Figure 3) showed that bud, SL and FL were clustered together but separate from ML, indicating the patterns of sequencing were changed according to the different tissues and position in the plant. To validate the accuracy of differential gene expressions, 12 genes in the terpene pathways were randomly selected and their expression levels were checked by qRT-PCR using specific primers as shown in Appendix A. As shown in Figure 2, the expression patterns of the 12 genes were in accordance with those of RNA-Seq data.

In plants, two biosynthetic pathways are responsible for the synthesis of IPP and DMAPP, the universal precursors of all terpenoids. Although the subcellular compartmentalization of the two pathways allows them to operate independently, metabolic cross-talk between the two pathways has been reported [22,23]. For the MVA pathway, 33 genes were identified and annotated as AACT, HMGS, HMGR, MVK, MPDC, PMK, IDI and FDS. IPP derived from the MVA pathway in the cytosol is further acted upon by IDI, a divalent, metal ion-requiring enzyme, to form dimethylallyl diphosphate (DMAPP). Then condensation of one DMAPP and two IPP molecules catalyzed by FDS leads to the formation of FPP(C_15_), the precursor of sesquiterpenes [24].

There are 20 genes in the MEP pathway, including DXS, DXR, MCT, CMK, MDS, HDS and HDR. This pathway leads to the formation of GPP (C_10_), the unique precursor for monoterpenes [24]. The MEP pathway was first elucidated in *Escherichia coli,* and plant homologs have been characterized using a combination of biochemical and genomic approaches [25]. Geranyl diphosphate synthase catalyzes the condensation of one molecule IPP and DMAPP to produce GPP (C_10_), the universal precursor of monoterpenes [24]. Then, an array of TPS catalyzes FPP and GPP to generate kinds of terpenoids. Taken all together, 29 genes were annotated as TPSs, including CIN/LIC, MYS/OCS, LIS/NES, NUDX, FAS, CPS/HUS, SQS and BAS, and among them, nerolidol synthase (CsNES) [17], *β*-ocimene synthase (CsOCS) [18] and linalool synthase (CsLIS) [19] were recently functionally characterized in *C. sinensis* plants.

Six enzymes were identified to generate monoterpenes, and 11 enzymes catalyze FPP to generate sesquiterpenes. Two farnesene synthase (FAS) was identified; the highest reads per kilobase per million (RPKM) value is for the bud, followed by FL and stem.

### 2.3. Integration of Gene Expression Levels and Terpenoid Profiles

To establish a genes to metabolites scenarios network of terpenes pathway, 75 TPS related genes (25 *MVA* genes, 24 *MEP* genes and 26 *TPS* genes) and 38 (of 41) detected terpenoid metabolites were applied for Pearson partial correlation analysis (r > 0.8 or r < −0.8, *p* < 0.05; Figure 4). The significant correlation networks were characterized by gene RPKM value to the metabolite accumulation patterns being either positively or negatively correlated in all these five tissues.

### 2.4. Expression of Three TPSs in Different Tissues and Their Response to MeJA

From the correlation network, five *TPSs (TPS1–TPS5)*, showed a positively correlated expression with *β*-farnesene content in *C. sinensis*. *TPS1* has been functionally verified and proven to act as a linalool synthase/nerolidol synthase [26]. In previous studies, methyl jasmonate (MeJA) has been applied as an elicitor in order to increase the volatiles, including farnesene, in strawberry [27], mango [28] and grapes [29]. Here, to detect the response of the five TPS to MeJA, 1 mM MeJA aqueous solution in 0.05% DMSO was sprayed to *C. sinensis* shoots, and expression level of these five TPSs were measured. As shown in Figure 5, *TPS2, TPS3* and *TPS5* strongly respond to MeJA (Figure 5a). Furthermore, the tissue-specific expression level of *TPS2, TPS3* and *TPS5* were also detected (Figure 5b). *TPS2* and *TPS5* are highly expressed in bud and FL, consistent with the accumulation of farnesene in *C. sinensis*. From the response of these 5 TPSs to MeJA and tissue-specific expression analysis, suggesting that *TPS2* and *TPS5* are good candidates related to farnesene biosynthesis.

### 2.5. TPS2 Catalyze α/β-Frnesene and β-Ocimene Formation In Vitro

The full-length *TPS2* and *TPS5* were cloned from *C. sinensis*, and phylogenetic analysis (Figure 5c) showed that *TPS2* and *TPS5* belong to TPS-a and TPS-c family, respectively. *TPS2* clustered together with CaFAS, which have been reported to be involved in the production of *α*-farnesene and *β*-ocimene in vitro but could only form *α*-farnesene in *C. sinensis* [20]. There is a farnesene synthase from *Vitis vinifera,* which also belongs to the TPS-a family [30].

To verify our gene-to-terpene network and to confirm whether *TPS2* is involved in the formation of farnesene, *TPS2* was cloned and expressed in *E. coli* Rosetta (DE3) cells. In vitro chemical products of the resulting recombinant proteins were analyzed by comparison to authentic standards. The main product produced by *TPS2* using FPP as a substrate was *β*-farnesene, followed by *α*-farnesene (Figure 6a). In contrast, *β*-ocimene is the resulting product when using GPP is used as the substrate. Together, our results indicate that *TPS2* is a multifunctional enzyme that could catalyze the formation of *α*- and *β*-farnesene, as well as *β*-ocimene in vitro.

### 2.6. Functional Analysis of TPS2 in Tea Plants

The in planta functions of *TPS2* were also studied in *C. sinensis* plants by transient suppression of the expression of *TPS2* using *TPS2* specific antisense oligonucleotides (AsODNs), according to Liu et al. [18]. A significant transcript level reduction in *TPS2* was detected after 24 h compared to control plants (CK; Figure 6b). Interestingly, only the abundance of *β*-ocimene and *α*-farnesene in the treated leaves were significantly reduced after AsODN-*TPS2* treatment (Figure 6c), indicate that *TPS2* plays a key role in the formation of both *β*-ocimene and *α*-farnesene. As *β*-farnesene is a very important and conserved alarm pheromone in response to aphids, its function in *β*-farnesene formation and aphids control need to be further studied.

## 3. Materials and Methods

### 3.1. Plant Materials and Chemicals

Five-year-old tea plants of *Camellia sinensis* cv. “Shu Cha Zao” grown at the experimental farm of Anhui Agricultural University in Hefei, China, were used in this study. Standards of geranyl diphosphate (GDP), farnesyl diphosphate (FDP), *β*-myrcene, *β*-ocimene, linalool, geraniol, *α*-citral, *β*-citral, methyl geranate, nerol, *D*-limonene, *trans*-linalool oxide (furanoid), neryl acetate, *L*-terpineol, *α*-terpinen, geranyl acetone, thymol, (*Z*)-nerolidol, *α*-farnesene, *β*-farnesene, (*E*)-nerolidol, humulene, *α*-ionone, *β*-ionone, (*E*)-γ-bisabolene, cadinol and caryophyllene were purchased from Sigma-Aldrich (St. Louis, MO, USA).

### 3.2. RNA Isolation and cDNA Library Construction

Total RNA was extracted from the bud, first leaf (FL), second leaf (SL), mature leaf (ML) and stem tissues of *Camellia sinensis* using Takara RNAzol reagent. Genomic DNA contamination was removed with DNaseI (Takara, Dalian, China). cDNA was obtained by reverse transcription from the total RNA using PrimeScript RT Reagent Kit (Code No. RR036A, Takara, Japan).

### 3.3. Transcript Assembly and Annotation

Raw sequence data (SRA accession: PRJNA491434) were processed by trimming the adaptor sequences and removing low-quality or empty reads. The clean reads were clustered using the Trinity v2.4.0 program (BroadInstitute, US, https://github.com/trinityrnaseq/trinityrnaseq). The resulting contigs were further assembled to produce unigene (http://seq.cs.iastate.edu/cap3.html). All these contigs were functionally annotated by searching the Uniprot database using the Blast program [31]. The level of gene expression in each sample was estimated by the number of reads that were assembled into a contig. The expression value was normalized for the length of the contig and for the total number of reads from a cDNA library by the reads per kilobase per million (RPKM). The gene expression data were then used for hierarchical clustering analysis of the samples and contigs using the TM4 MeV software package.

### 3.4. GC–MS Analysis of Volatile Compounds in Samples

For volatile compound analysis in *Camellia sinensis* tea samples, solid-phase microextraction (SPME) was introduced, frozen samples were ground to a fine powder in liquid nitrogen, and transferred to a 20 mL headspace bottle. Two microliters of ethyl caprate (58.84 μL/mL in water) were used as an internal standard to the sample. Samples were incubated at 65 °C for 50 min. Metabolite analysis was performed by gas chromatography (Agilent 7890A)/mass spectrometry (Agilent 5975C; GC/MS), with a DB-WAX capillary column (30 m × 0.25 mm × 0.25 μm; Agilent) to separate metabolites. Pure helium was used as the carrier gas, with a flow rate of 1 mL/min. The GC oven was maintained at 40 °C for 5 min and a ramp of 5 °C/min to 250 °C for 5 min under splitless full-scan mode. The energy was −70 eV in electron impact mode. The mass spectrometry data were acquired in full-scan mode with an m/z range of 20–400 after a solvent delay of 0 s. All metabolite peaks were recorded and stored in the format of ChemiStation data files. To estimate the retention index values, a mixture of even-numbered chain n-alkenes (C10–C40, Restek, FL, USA, catalog no. 31266) as a standard was used to deconvolute the metabolite peaks. At least six biological repetitions were performed for each sample.

### 3.5. Transcript Expression Analysis of Genes Involved in Terpenes

Quantitative PCR (Q-PCR) assays were performed on a CFX96 platform (Bio-Rad, California, CA, US), using specific primers Appendix A, and the Top Green Q-PCR SuperMix (TransGen, Beijing, China) according to the manufacturer’s instructions. PCR reaction efficiencies for all test genes were more than 90%, and their transcript levels were calculated using the 2^–ΔΔ*C*T^ method [32].

### 3.6. Integration of Gene Expression and Terpenoid Molecule Profiles

Correlation between the expression levels of 82 genes and the profiles of 41 terpenoids in the bud, FL, SL, ML and stem were carried out using the program R 2.10.1. RPKM values for genes and the peak values of metabolites were used as a matrix for Pearson partial correlation analysis, r > 0.8 or r < −0.8, *p* < 0.05. The resulting correlation networks were obtained and visualized using Cytoscape software (Cytoscape 2.6.3). The phylogenetic tree was constructed by the neighbor-joining computation method. The evolutionary distances were computed using the *p*-distance method and are in the units of the number of amino acid differences per site. The analysis involved 23 amino acid sequences. All positions containing gaps and missing data were eliminated. There was a total of 388 positions in the final dataset. Evolutionary analyses were conducted in MEGA7 [26].

### 3.7. Methyl Jasmonate Treatments

To examine the effects of Methyl jasmonate (MeJA, Sigma Aldrich) on the expression of the four selected farnesene correlated genes, a 1 mM MeJA aqueous solution in 0.05% DMSO was sprayed every hour as a fine mist onto tea shoots, leaves were collected at 0, and 12 h respectively.

### 3.8. Gene Cloning and Prokaryotic Expression

To get the full length of *TPS2*, the primers (forward: AATGCCAAAG GAGGTCAGCC; reverse: TTGAGTTGCAAGAGACTGTAGAAGA) and Prime -STAR MAS DNA Polymerase (Takara, Dalian, China) were used for cloning according to the manufacturer’s instructions. The PCR conditions were shown as follows: 3 min at 98 °C, 35 cycles of 98 °C for 10 s, 60 °C for 5 s, 72 °C for 1 min and a final extension at 72 °C for 10 min. The PCR products were gel purified using a Gel Extraction Kit (Qiagen, Hilden, Germany), cloned into the pGEX-4T-1 vector (Promega, Wisconsin, USA), and then transformed into Trans-1t1 competent cells for sequencing. The resulting chimeric gene was expressed in *Escherichia coli* BL21 (DE3) Rosetta and incubated at 37 °C until OD_600_ = 0.6–0.8, followed by 1 mm IPTG induction at 16 °C for 22 h. Following IPTG induction, the cells were harvested at 4000× *g* for 10 min, then the expressed protein was isolated and refolded as described previously [33]. Proteins were separated by electrophoresis on a 12% denaturing polyacrylamide gel and stained with 0.25% Coomassie Blue R-250 for a purity check. The partially purified protein solutions were stored at −20 °C for later use.

### 3.9. Enzyme Assay

Enzyme activity assays were carried out in 1 mL reaction buffer (pH 7.5 50 mM Tris-HCL, 10 mM MgCl2, and 5 mM DTT) containing 50–100 μg of purified recombinant protein and 5 μg FPP/GPP substrate in a 50 mL glass tube. The mixture was incubated at 30 °C for 1 h and then at 42 °C for 15 min [17]. After incubation, the volatiles collected by the solid-phase microextraction (SPME) fiber (2 cm-50/30 μm DVB/CarboxenTM/PDMS Stable FlexTM from Supelco) were subjected to GC–MS analysis described as above.

### 3.10. TPS2 Enzyme Assay in Camellia Sinensis

The functional assay of TPS2 in tea plants was carried out by gene suppression of *TPS2* in *C. sinensis,* according to Liu et al. [18]. Candidate sequences of the antisense oligonucleotides (AsODN-TPS2: TTGTTGATCTGCTAGCATTC) with complementarity to the segment of *TPS2* were selected using Soligo software (http://sfold.wadsworth.org/cgi-bin/index.pl) with *TPS2* sequences as inputs. AsODNs were synthesized by General Biological System (Anhui) Co., Ltd (Anhui, China). For validation of target gene suppression using AsODN, naturally growing tender *C. sinensis* shoots with three leaves were excised and placed in Eppendorf tubes containing 1 mL of 20 μm AsODN-*TPS2* (to suppress *TPS2*) for 24 h. All leaves were harvested and used for volatile extraction and terpene analysis as described above. Three replicates were employed for each sample.

## 4. Conclusions

A gene-to-terpene network was revealed by an integrated approach combining metabolites (41 terpenes), target gene transcripts (82 genes), and gene function analyses in *C. sinensis*. Five terpene synthases whose expression positively correlated with farnesene were selected. Both in vitro and in planta analysis confirmed that TPS2 is involved in the biosynthesis *α*-farnesene, *β*-ocimene and *β*-farnesene in *C. sinensis* plants. These data also suggested that our gene-to-terpene network was a reliable method for novel terpene-related genes discovery in *C. sinensis* and other economical plants

## Figures and Tables

**Figure 1 ijms-21-00655-f001:**
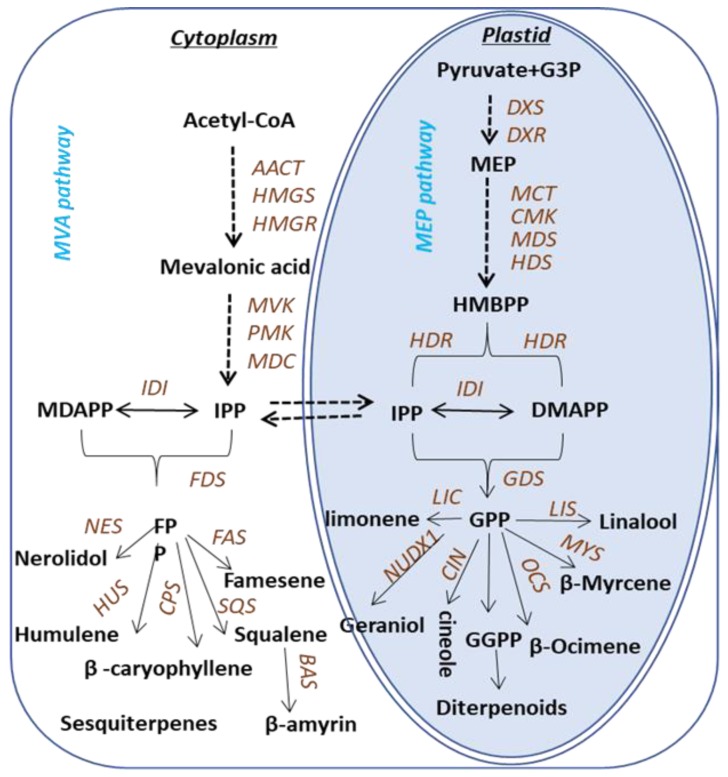
Pathway associated with terpene synthesis in *Camellia sinensis.* The following enzymes and compounds are involved in the MVA pathway: AACT, acetoacetyl-CoA thiolase; HMGS, 3-hydroxy-3-methylglutaryl- coenzyme (a synthase); HMGR, 3-hydroxy-3-methylglutaryl coenzyme (a reductase); MVK, mevalonate kinase; PMK, 5-phosphatemevalonate kinase; MDC, mevalonate-5-phosphate decarboxylase; IPP, isopentenyl pyrophosphate; IDI, isopentenyl pyrophosphate isomerase; DMAPP, dimethylallyl pyrophosphate. The following enzymes and compounds are involved in the MEP pathway: DXS, 1-deoxy-d-xylulose-5-phosphate synthase; DXR, 1-deoxy-d-xylulose5-phosphate reductase; MCT, 2-C-methyl-d-erythritol-4-phosphate cytidyltransferase; CMK, cytidyl (4-diphospho)-2-C-methyl-d-erythritol kinase; MDS, 2-C-methyl-d-erythritol-2,4-cyclodiphohate synthase; HDS, 1-hydroxyl-2-methyl-2-(*E*)-butenyl-diphosphate synthase; HDR, 4-hydroxy-3-methylbut-2-enyldiphosphate reductase; IDI, isopentenyl pyro-phosphate isomerase. The following enzymes are involved in the sesquiterpenes biosynthetic pathway: FDS, farnesyl diphosphate; FPP, farnesyl pyrophosphate; FAS, farnesene synthase; HUS, humulene synthase; NES, nerolidol synthase; SQS, squalene synthase. The following enzymes are involved in the pathways toward monoterpene and diterpene formation: GDS, Geranyl diphosphate; GPP, Geranyl pyrophosphate; LIS, linalool synthase; LS, limonene synthase; MYS, β-Myrcene synthase; OCS, β-Ocimene synthase; NUDX1, Nudix hydrolase 1.

**Figure 2 ijms-21-00655-f002:**
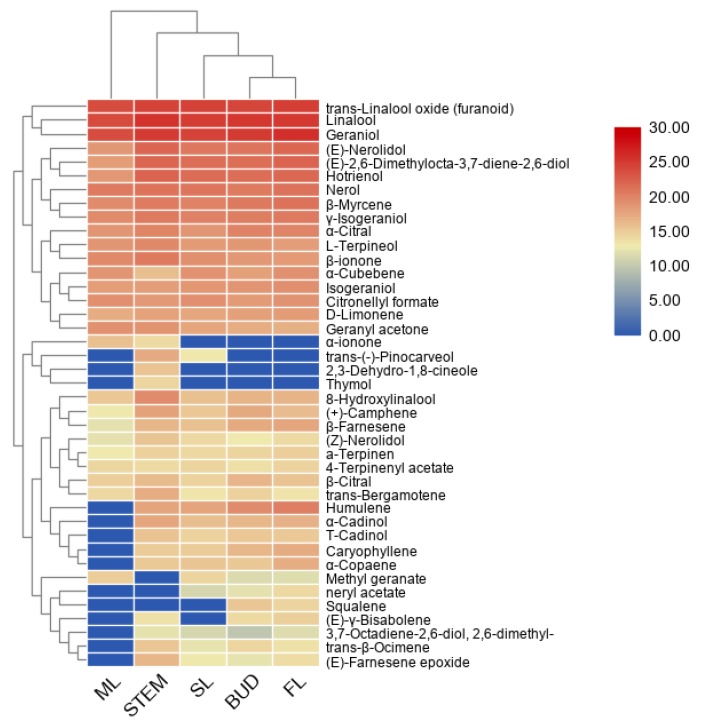
A heatmap coupled with clustering analysis for 41 terpenes detected in five tissues of tea plants. ML, mature leaf; SL, the second leaf; FL, the first leaf.

**Figure 3 ijms-21-00655-f003:**
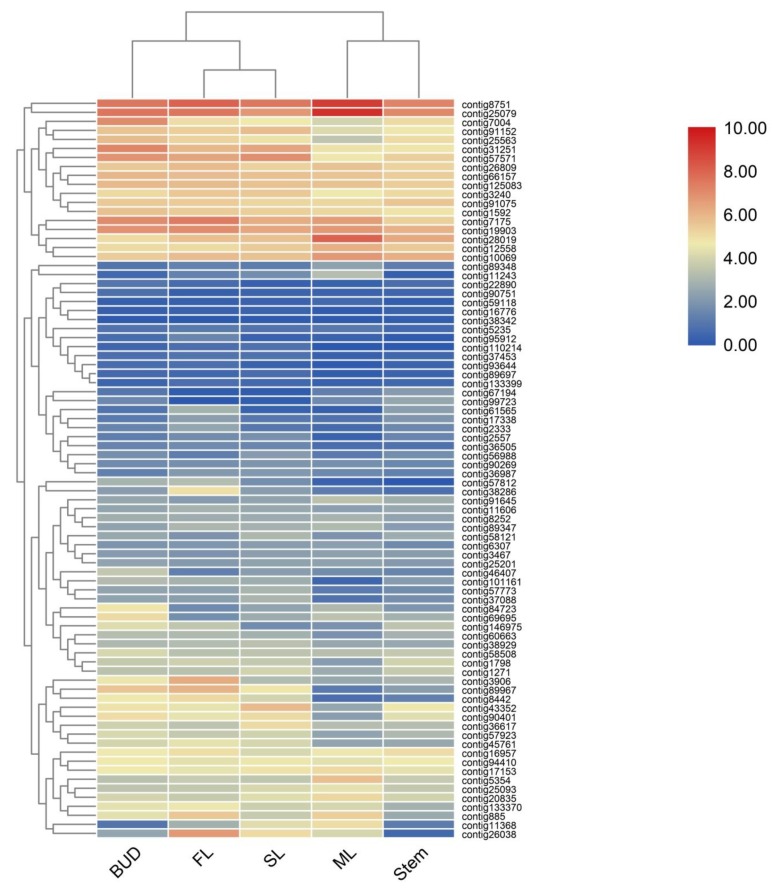
A heatmap coupled with clustering analysis of 82 terpenes related genes in five tissues of tea plants. ML, mature leaf; SL, the second leaf; FL, the first leaf.

**Figure 4 ijms-21-00655-f004:**
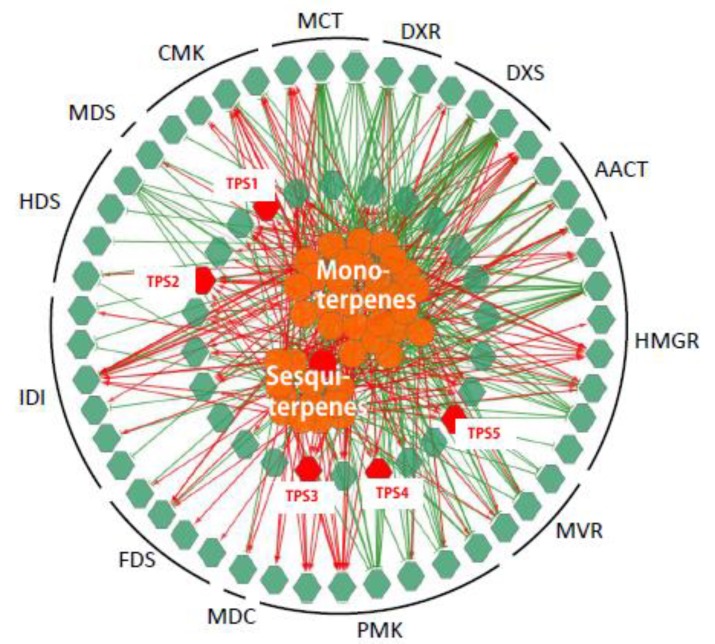
Correlation networks generated by the integration of expression levels of genes and terpenoids accumulation. The networks were generated by the integration of the expression levels (reads per kilobase per million (RPKM) value) of 75 (of 82) genes (green hexagons) and the accumulation of 41 terpenoids (orange circulars) using Pearson partial correlation analysis (r > 0.8 or r < −0.8, *p* < 0.05). Red hexagons, genes related to farnesene, named *TPS1-5*. Red links, positive correlation (r > 0.8): green links, negative links (r < −0.8).

**Figure 5 ijms-21-00655-f005:**
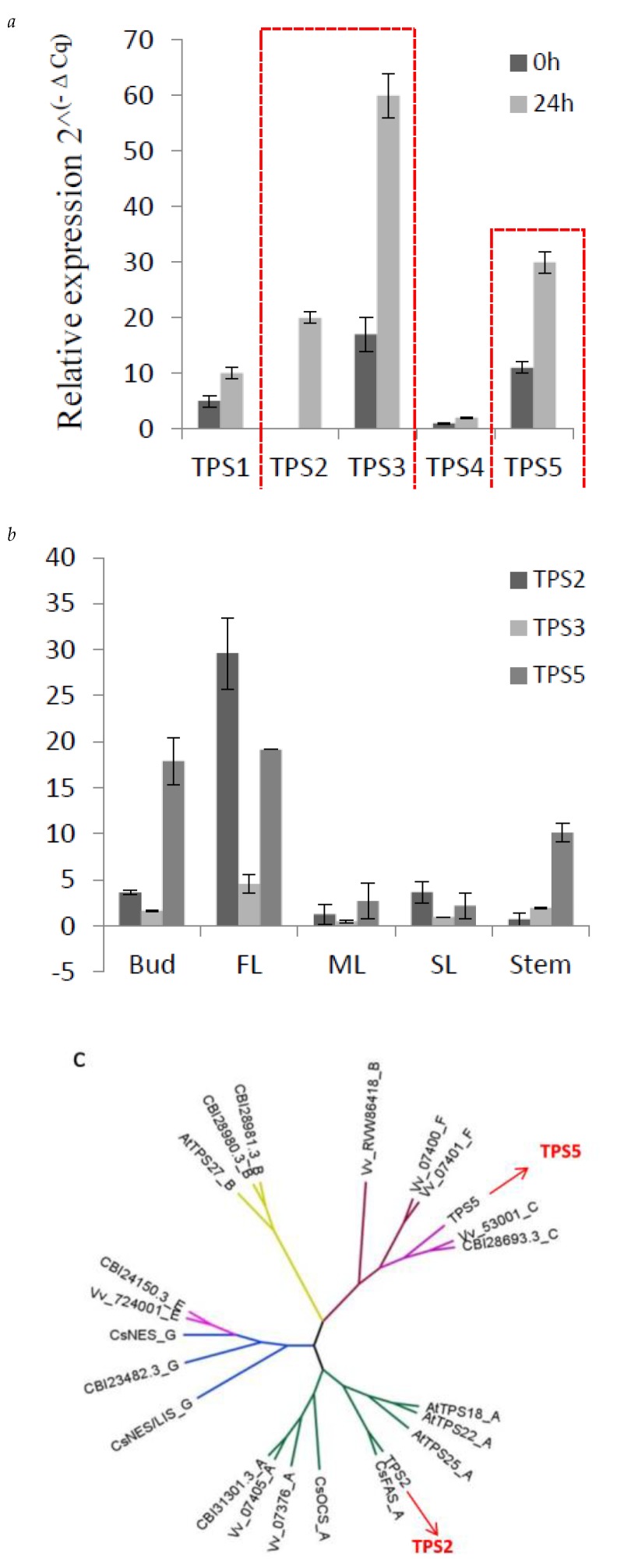
Expression of three *TPSs* in different tissues and their response to MeJA. (**a**) q-PCR results of the expression level of *TPS1-TPS5* responds to MeJA, red line to highlight the changes of *TPS2, TPS3* and *TPS5*. (**b**) q-PCR showing the different expression levels of *TPS2, 3* and *5* in five tissues, (**c**) the phylogenetic tree of *TPS2* and *TPS5,* which was obtained by using MEGA-7 software with the neighbor-joining method. At, *Arabidopsis thaliana*; Vv, *Vitis vinifera*; CsAFS, *α*-farnesene synthase in *Camellia sinensis; CsOCS,*
*β*-ocimene synthase in *Camellia sinensis*; CsNES, nerolidol synthase in *Camellia sinensis;* CsNES/LIS, nerolidol/linalool synthase in *Camellia sinensis.*

**Figure 6 ijms-21-00655-f006:**
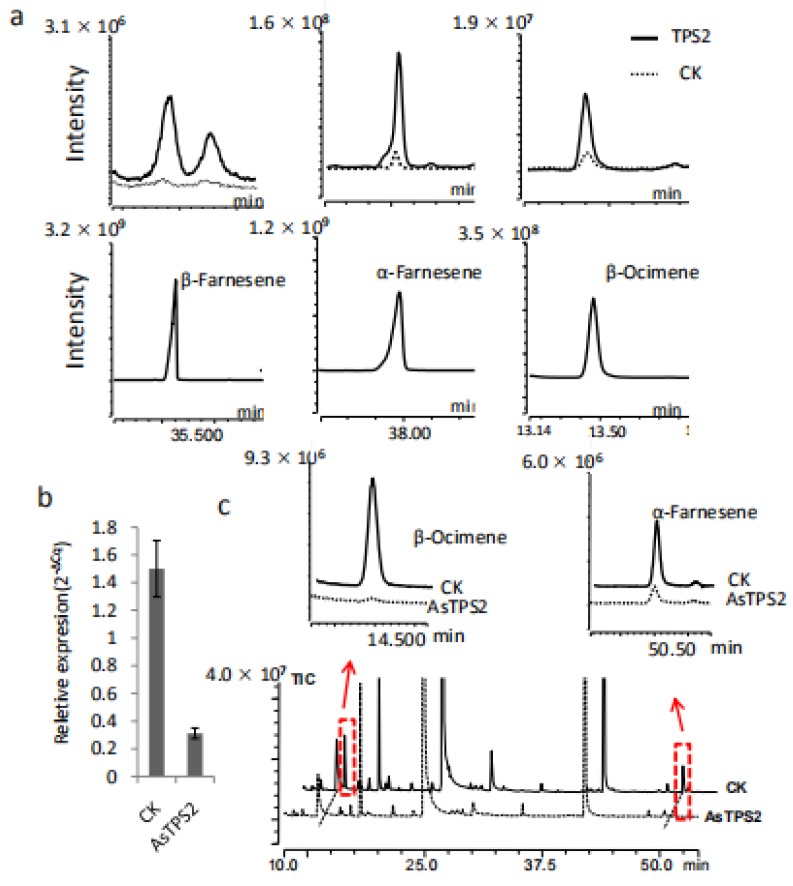
Functional characterization of *TPS2* in vitro and in plants. (**a**) *β*-Farnesene, *α*-Farnesene and *β*-Ocimene produced by *TPS2* using FPP and GPP as substrates respectively. (**b**) The expression level of *TPS2* was reduced after treated with AsODN-*TPS2* and change in the abundance of *α*-Farnesene and *β*-Ocimene in tea treated with AsODN-*TPS2* (highlighted by the red boxes and arrows) (**c**).

**Table 1 ijms-21-00655-t001:** Monoterpenoids detected in different tissues of tea plants.

Monoterpenes(26)	Chemical Formulas	Tissues	Types and Number
*β*-Myrcene *	C10H16	All tissues	Acyclic(17)
*trans-β*-Ocimene *	C10H16	Bud, FL, SL, Stem	
Linalool *	C10H18O	All tissues	
Geraniol *	C10H18O	All tissues	
*α*-Citral *	C10H16O	All tissues	
*β*-Citral *	C10H16O	All tissues	
Methyl geranate *	C11H18O2	ML, SL	
Nerol *	C10H18O	All tissues	
8-Hydroxylinalool	C10H18O2	All tissues	
3,7-Octadiene-2,6-di-ol, 2,6-dimethyl	C10H18O2	FL, SL, Stem	
neryl acetate *	C12H20O2	FL	
Isogeraniol	C10H18O	All tissues	
*γ*-Isogeraniol	C10H18O	All tissues	
Geranyl acetone *	C13H22O	All tissues	
Citronellyl formate	C11H20O2	All tissues	
DMDD	C10H18O2	All tissues	
Hotrienol	C10H16O	All tissues	
*D*-Limonene *	C10H16	All tissues	Monocyclic(6)
*trans*-Linalool oxide (furanoid) *	C10H18O2	All tissues	
*L*-Terpineol *	C10H18O	All tissues	
*α*-Terpinen *	C10H16	All tissues	
4-Terpinenyl acetate	C12H20O2	All tissues	
Thymol *	C10H14O	Stem	
2,3-Dehydro-1,8-cineole	C10H16O	Stem	Bicyclic(3)
(+)-Camphene	C10H16	All tissues	
*trans*-(−)-Pinocarveo	C10H16O	Stem	

Note: DMDD: (*E*)-2,6-Dimethylocta-3,7-diene-2,6-diol. * were identified using authentic standard compounds.

**Table 2 ijms-21-00655-t002:** Sesquiterpenes and triterpenes detected in tea plants.

Sesquiterpenes(14)	Chemical Formulas	Tissues	Types and Number
*(Z)*-Nerolidol *	C15H26O	All tissues	Acyclic(3)
*α*-Farnesene *	C15H24	All tissues	
*(E)*-Nerolidol *	C15H26O	All tissues	
*(E)*-Farnesene epoxide *	C15H24O	Bud, FL, SL, Stem	Monocyclic(5)
Humulene *	C15H24	Bud, FL, SL, Stem	
*α*-ionone *	C13H20O	ML, Stem	
*β*-ionone *	C13H20O	All tissues	
*(E)-γ*-Bisabolene *	C15H24	Bud, FL, Stem	Bicyclic(4)
*trans*-Bergamotene	C15H24	All tissues	
*α*-Cadinol	C15H26O	Bud, FL, SL, Stem	
*T*-Cadinol *	C15H26O	Bud, FL, SL, Stem	
Caryophyllene *	C15H24	Bud, FL, SL, Stem	
*α*-Cubebene	C15H24	All tissues	Tricyclic(2)
*α*-Copaene	C15H24	Bud, FL, SL, Stem	
Squalene	C30H50	Bud, FL	Triterpene(1)

Note: * were identified using authentic standard compounds.

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
