# Peer review of "Scenarios of Genes-to-Terpenoids Network Led to the Identification of a Novel α/β-Farnesene/β-Ocimene Synthase in Camellia sinensis"

_ijms, 2020, doi:10.3390/ijms21020655_

Round 1
Reviewer 1 Report
Dear Authors,
The article presents quite clearly the idea and experiment. However, some minor changes should be introduced, namely:
=== MAIN ===
Results
The involved strategies/methods could be introduced briefly.
Summary
In summary, propose direct and far applications of your discovery. If possible, add a reference example of an analogic discovery that resulted in a real application.
=== EDITORIAL ===
Serious language correction is needed. Check the homogeneity of American/British style. See, what was pointed in the Abstract only:
l.10
'a vital roles' - > 'a vital role' OR 'vital roles'
typos:
l.10
'performace'
l.14
'sesuiterpenes'
l.14
'forty one' -> 'forty-one'
l.17
'positive' -> 'positively'
l.18
'biosynthsis', 'imprtant'
l.19
'enzymtic'
l.20
'anasysis'
===
'gene-to-terpene' -> Use only one form with a consequence in whole work (e.g., with hyphens).
Supply missing commas.
Latin inclusions (via, in vitro, in planta), and some descriptors (m/z, (E)- ) -> Chenge to italics.
Change all 'um' units to 'μm' where needed; do the same with 'ug' and 'uL'.
===
l.28
'nutritional properties' -> use more adequate term; digestive carbohydrates, fat, proteins are 'nutritional'
fig.1, l.68
e.g., '(Latin)b-caryophyllene', '(Greek)b-Ocimene', 'Beta-amyrin' -> Use one form of descriptors with a consequence.
Instead of a very long caption, consider to define abbreviations alphabetically, right to the picture, with petit font; there is enough place.
l.92
When introducing any producer, use the address of headquarter, not of the section
tab. 1&2, fig. 1&2
Take care of the names/descriptors unification (Greek)- or (Latin)-, as above.
===
Table 1,2
Prepare and set in editable form, according to the MDPI policy and template.
Btw. check the names and use unambiguous forms, e.g.,
Lonone -> ionone
(?)Nerolidol and (E)-Nerolidol -> is the first (Z)-?
a-cadinol and (?)-cadinol
?citral and b-citral
…
===
References
Check the MDPI/IJMS journal policy and apply
Check if all abbreviated names are given in section Abbreviations (also those introduced with full names in the text). Arrange them in alphabetical order.
Author Response
The article presents quite clearly the idea and experiment. However, some minor changes should be introduced, namely:
Answer: Thank you for your positive evaluation of our manuscript. We have carefully checked your helpful remarks and suggestions to make our manuscript more precise and rigor.
=== MAIN ===
Results
The involved strategies/methods could be introduced briefly.
Answer: Thank you for your suggestion, we have simplified some part of results involved in methods, please refer to line 178 “the volatiles in different tissues including, the bud, the first leaf … and stem of tea” simplified as ”five different tissues of tea”; line 252-254 “The significant correlation networks … in all these five tissues. ” have been deleted. And line 304-306 have been deleted because the gene silencing strategies have been mentioned in the methods part.
Summary
In summary, propose direct and far applications of your discovery. If possible, add a reference example of an analogic discovery that resulted in a real application.
Answer: Thank you for your suggestion, we have added some applications of our finding in the summary, please refer to line 317-321.
=== EDITORIAL ===
Serious language correction is needed. Check the homogeneity of American/British style.
Answer:The manuscript has been carefully revised by us and by a language polishing company (EditorBar, Beijing, China) as suggestions to improve our manuscript.
See, what was pointed in the Abstract only:
l.10
'a vital roles' - > 'a vital role' OR 'vital roles'
Answer: Thank you, we have revised it.
typos:
l.10
'performace'
Answer: Thank you, we have revised it.
l.14
'sesuiterpenes'
Answer: Thank you, we have revised it.
l.14
'forty one' -> 'forty-one'
Answer: Thank you, we have revised it.
l.17
'positive' -> 'positively'
Answer: Thank you, we have revised it.
l.18
'biosynthsis', 'imprtant'
Answer: Thank you, we have revised it.
l.19
'enzymtic'
Answer: Thank you, we have revised it.
l.20
'anasysis'
Answer: Thank you, we have revised it.
===
'gene-to-terpene' -> Use only one form with a consequence in whole work (e.g., with hyphens).
Supply missing commas.
Answer: Thank you very much for your suggestion. We have revised and keep “gene-to-terpene” throughout the manuscript as suggested.
Latin inclusions (via, in vitro, in planta), and some descriptors (m/z, (E)- ) -> Chenge to italics.
Answer: Thank you. We have revised it carefully throughout the manuscript.
===
l.28
'nutritional properties' -> use more adequate term; digestive carbohydrates, fat, proteins are 'nutritional'
Answer: Thank you for your suggestion, we have changed it to “health benefits”.
fig.1, l.68
e.g., '(Latin)b-caryophyllene', '(Greek)b-Ocimene', 'Beta-amyrin' -> Use one form of descriptors with a consequence.
Answer: Thank you for your suggestion, we have carefully revised them in the Figure 1 and upload the new version.
Instead of a very long caption, consider to define abbreviations alphabetically, right to the picture, with petit font; there is enough place.
Answer: Thank you for your suggestion, we have tried to put the abbreviations alphabetically right to the picture, however, we cannot make it easier as so many abbreviations here, and the space is not enough. We thanks for your understanding.
l.92
When introducing any producer, use the address of headquarter, not of the section
Answer: Thank you for your suggestion, we have revised it.
tab. 1&2, fig. 1&2
Take care of the names/descriptors unification (Greek)- or (Latin)-, as above
Answer: Thank you for your suggestion, we have revised it
===
Table 1,2
Prepare and set in editable form, according to the MDPI policy and template.
Btw. check the names and use unambiguous forms, e.g.,
Lonone -> ionone
(?)Nerolidol and (E)-Nerolidol -> is the first (Z)-?
a-cadinol and (?)-cadinol
?citral and b-citral
Answer: Thank you for your suggestion, we have uploaded the editable, the names and descriptors have been corrected as suggested.
===
References
Check the MDPI/IJMS journal policy and apply
Answer: Thank you for your suggestions. We have revised the format of the references according to the IJMS journal policy.
Check if all abbreviated names are given in section Abbreviations (also those introduced with full names in the text). Arrange them in alphabetical order.
Answer: Thank you for your suggestion, we have checked the Abbreviations part to make sure all abbreviated names have been included.
Reviewer 2 Report
The authors of ijms-690833 «Scenarios of genes-to-terpenoids network led to the identification of a novel α/β-farnesene/β-ocimene synthase in Camellia sinensis» present new results of GC-MS-based metabolomics and RNAseq transcriptomics data combined to propose gene-to-metabolite network applied to terpenes biosynthesis allowing the identification of a new terpene synthase with a substrate specificity toward the formation of alpha and beta farnesene as well as beta ocimene in tea.
The introduction provide sufficient background and include all relevant references, the research design appropriate, the methods adequately described, the results clearly presented and the conclusions supported by the results.
I simply observed some minor things to fix before acceptation of this work:
1- M&M, paragraph 2.2. Sequencing used is not mentioned.
2- Figure 2 legend. Add the abbreviations used for the different tissue examined in the figure legend too.
3- Figure 4 legend. Indicate the color used for the links (red or green) corresponding to the correlation r>0.8 and r<-0.8.
4- Figure 5. TPS5 “5” is missing on the figure.
5- There is some typing errors within the document (eg Abstract line 19 “biosynthesis”, “e” is missing). Please check the whole document.
Author Response
The authors of ijms-690833 «Scenarios of genes-to-terpenoids network led to the identification of a novel α/β-farnesene/β-ocimene synthase in Camellia sinensis» present new results of GC-MS-based metabolomics and RNAseq transcriptomics data combined to propose gene-to-metabolite network applied to terpenes biosynthesis allowing the identification of a new terpene synthase with a substrate specificity toward the formation of alpha and beta farnesene as well as beta ocimene in tea.
The introduction provide sufficient background and include all relevant references, the research design appropriate, the methods adequately described, the results clearly presented and the conclusions supported by the results.
Answer: Thank you for your positive evaluation of our manuscript. We have carefully checked your helpful remarks and suggestions to make our manuscript more precise and rigor.
I simply observed some minor things to fix before acceptation of this work:
1- M&M, paragraph 2.2. Sequencing used is not mentioned.
Answer: Thank you for your suggestion. We have added the sequencing in the method part.
2- Figure 2 legend. Add the abbreviations used for the different tissue examined in the figure legend too.
Answer: Thank you for your suggestion, we have added the abbreviations as suggested.
3- Figure 4 legend. Indicate the color used for the links (red or green) corresponding to the correlation r>0.8 and r<-0.8.
Answer: Thank you for your suggestion, we have made it clear in the Figure 4 legend.
4- Figure 5. TPS5 “5” is missing on the figure.
Answer: Thank you for your kindly reminding, we have revised it.
5- There is some typing errors within the document (eg Abstract line 19 “biosynthesis”, “e” is missing). Please check the whole document.
Answer: Thank you for your suggestion. We have carefully revised our manuscript as suggested.